# The Natural Philosophy of Economic Information: Autonomous Agents and Physiosemiosis

**DOI:** 10.3390/e23030277

**Published:** 2021-02-25

**Authors:** Carsten Herrmann-Pillath

**Affiliations:** Max Weber Centre for Advanced Cultural and Social Studies, Erfurt University, 99089 Erfurt, Germany; carsten.herrmann-pillath@uni-erfurt.de

**Keywords:** semantic information, economic agency, physiosemiosis, Peirce, thermodynamics of information, evolution, final causes

## Abstract

Information is a core concept in modern economics, yet its definition and empirical specification is elusive. One reason is the intellectual grip of the Shannon paradigm which marginalizes semantic information. However, a precise concept of economic information must be based on a theory of semantics, since what counts economically is the meaning, function and use of information. This paper introduces a new principled approach to information that adopts the paradigm of biosemiotics, rooted in the philosophy of Charles S. Peirce and builds on recent developments of the thermodynamics of information. Information processing by autonomous agents, defined as autopoietic heat engines, is conceived as physiosemiosis operating according to fundamental thermodynamic principles of information processing, as elucidated in recent work by Kolchinsky and Wolpert (KW). I plug the KW approach into a basic conceptual model of physiosemiosis and present an evolutionary interpretation. This approach has far-reaching implications for economics, such as suggesting an evolutionary view of the economic agent, choice and behavior, which is informed by applications of statistical thermodynamics on the brain.

## 1. Introduction

This paper explores a semantic theory of information grounded in thermodynamics. The motivation is to clarify the role of information in economics, with a focus on the microfoundations: I will not explore wider dimensions on the macro level, such as the knowledge economy, digitalization or growing importance of intangible goods in economic structure. The relationship between thermodynamics and economics has been productive but is also fraught with conceptual difficulties. It has so far been almost exclusively developed in ecological economics [1], trailing the breakthrough contributions by Nicholas Georgescu-Roegen [2]. However, Georgescu-Roegen was also deeply influential in blocking explorations on information, as he erroneously battled against the Boltzmann approach to entropy [3]. Today, given recent advances in the thermodynamics of information, we have the opportunity to combine energetic aspects emphasized by Georgescu-Roegen and the thermodynamics of information [4], and to inquire whether such a synthesis has also consequences for the concept of information in economics. My paper explores generic aspects of information, and not economic information specifically. However, as we will see, the generic aspects include many considerations which have direct implications for economics, of which I explore one aspect in the final section—the theory of economic agency. 

In modern economics, the concept of information has obtained a central position, with Hayek’s [5] breakthrough contribution on the “use of knowledge in society” pointing towards an integration of the micro and the macro level. Yet, and strangely, it mostly remains vaguely defined as far as the microfoundations are concerned. For example, one of the founders of modern information economics, Joseph Stiglitz, does not provide a definition in his 2017 overview of the field [6]. As amply demonstrated in the intellectual history of the term by Mirowski and Nik-Kah [7], despite its centrality the concept seems elusive and packed with many hidden ontological assumptions which must be put to closer scrutiny. Precise definitions which inform general economics can be found in game theory, where information refers to a set of possible states of the world and a function that partitions these states into what constitutes an agents’ knowledge [8], p. 67ff. This view is inspired by the Shannon concept of information: The “world” appears like the sender of messages, and the economic agent is the receiver who decodes the messages. The meaning of the game is covered by the construct of “common knowledge” which frames the agents’ strategies.

However, economic uses of information often go beyond this formal definition. In particular, since the inception of the information economics paradigm, information has been approached as a kind of “thing”, that is, as an economic good with extensive property. This corresponds to the Shannon view, yet there is no clear conception of what information constitutes materially on the micro level. In the wake of the IT revolution, this view has been bolstered by the immense economic significance of “data” in newly emerging technologies and business models, which highlights the quantitative aspects and appears to suggest a clear material interpretation. Interestingly, this development has heightened interest in the energetic aspects of information, for mere practical reasons [9]. However, so far these concerns have not inspired foundational work on energy and information in general, i.e., beyond the specific setting of digital technologies which literally embody Shannon information.

The pull of the Shannon concept of information is so strong that even most audacious attempts at introducing new perspectives eventually stay within its confines. Interestingly, two contributions by physicists loom large here. In his 1994 book, Robert Ayres [10] clearly diagnosed the need to distinguish Shannon information from another aspect of information which he called “survival relevant information” (SR information), thus directly referring to an evolutionary framework. Shannon information is “distinguishability information” (D information), which refers to the quantity of information that is embodied in a pattern referred to a given state space. SR information is information that is functional for sustaining the existence of that pattern. For example, we can measure the information content of a DNA molecule in two ways: one is to fix a state space for the structures and elements of the molecule, and hence determine its negentropy relative to that state space (D information), the other is to assign information value to only those parts of the molecule that fulfil a function in biological processes (SR information). However, Ayres failed in further sharpening the concept of SR information, and in his later work on energy and economy [11] he blanked out these more fundamental issues, thus finally dissociating his physical approach to economic growth from developing a physical theory of economic information that, after all, is the ultimate driver of growth, such as market intelligence or technological know-how: that is, he focused on the macro level and sidelined microfoundations. Similarly, in his 2015 book, Hidalgo [12] starts out from developing a physical view on information informed by thermodynamics but then falls back on the notion of complexity as determined by the Shannon measures. His measure of economic complexity in international trade [13] only catches the mere quantitative aspects of a state space of goods and technologies but does not cover the functions of those goods in the economic process.

In staying close to the Shannon concept of information and side-lining a more precise economic definition of information, economics fails to develop a theory of semantic information [14]. To be fair, this is not Shannon’s fault, who explicitly excluded semantic information in formulating his theory. When considering economic processes, the semantic dimension is essential because this refers to the use of information. This aspect is also fundamental for the general concept of information: Information is always relative to an observer for whom the information is valuable, in the sense of fulfilling a certain function, and information is about something, i.e., a referent [15]. In this paper, I suggest specifying the concept of semantic information by referring to Charles S. Peirce’s theory of signs [16], as deployed in modern biosemiotics [17,18]. This view allows one to approach the materiality of information in a comprehensive and systematic way, thus providing the basis for a physical view on information in economics: I refer to this as “physiosemiotics” [19]. 

The paper continues with developing the physiosemiotic view as a necessary complement to the Shannon concept of information. The physiosemiotic view essentially refers to an entity that interprets and uses information. I argue that the most general conception of that entity is that of an “autonomous agent” as introduced by Stuart Kauffman [20]. The autonomous agent is a physical entity which is capable of maintaining thermodynamic work cycles to sustain its existence as defined by internally determined goals. Fundamental economic notions such as “growth” and “efficiency” can be already meaningfully introduced at this step. In the next section, I sketch the important contribution by Kolchinsky and Wolpert [21] on a physical approach to semantic information, which allows one to formulate the basic economics of information grounded in thermodynamics. Going beyond Kolchinsky and Wolpert, I claim that this view can translated into the conceptual frame of Peircean semiotics. I conclude by sketching an important and exemplary consequence for economics proper, conceptualizing economic agents as autonomous agents.

## 2. Basic principles of the Physiosemiotics

### 2.1. Shannon Information versus Semiotics

Peirce’s theory of signs differs fundamentally from Shannon’s approach in being triadic, and not dyadic: Shannon information only distinguishes between sender and receiver, whereas semiotics introduces a “third”—the sign (more, precisely, “sign vehicle” or “representamen” in some distinctly Peircean terminology) (the following discussion refers to Figure 1) [22]. Based on this, semiotics makes the role of inference as a mechanism of gaining information from interpreting signs explicit, which is taken for granted in the Shannon approach (deliberately so, as being irrelevant for the question posed, i.e., transmission of information [23], p. 160ff). Finally, semiotics reassesses the roles of sender and receiver. As salient in the original use of Shannon information, the sender is the one who creates the information to be transmitted via the channel—the “message”. The message may be coded, and the receiver only needs to know the code used by the sender to retrieve the complete information, unless there is “noise” or limited channel capacity partly impeding complete transmission. In contrast, semiotics emphasizes the role of the receiver in creating the information content of the message: if the receiver already knows it, there would be no information at all, or if the information may be new, but is useless, it assumes an informational value of zero. In semiotics, information is mainly with the receiver, commonly denoted “interpretant” (although, as we shall see, the full picture remains triadic). 

There is another, deeper distinction on the ontological level. What is the physical nature of information in the two frames of reference? In the Shannon case, physics comes into play when considering the interaction between sender and receiver via the channel, which is a purely physical process, including the message as a physical entity (such as electromagnetic waves). However, the state space in which the information content (“bits”) is defined is non-physical, since it is arbitrarily defined by the code employed by the sender and shared with the receiver. Although that may be physically embodied, it is not physically determined: the code may be embodied in physical structures such as the neurobiology of sender and receiver, but even then there is no physical explanation of that code, i.e., its embodied meaning. 

This distinction becomes important when considering the relationship between Shannon information and physical entropy. In the Shannon approach, physical entropy only comes into play when we consider “noise” which is a physical phenomenon that impacts the physical transmission process. In terms of causal notions, there is an analytical distinction between the information as message, which is non-physical, and the transmission process; this appears to be another form of Cartesian dualism of mind and matter, thus suggesting that “information” is disembodied and hence can be conjugated with arbitrarily many forms of physical manifestations. This Cartesian dualism has deeply shaped the concept of information in economics. 

In comparison, the triadic view of physiosemiotics allows one to distinguish between the physical nature of the causal interaction between sender and receiver, the physical nature of the sign and the physical nature of the consequences that define the value of information, i.e., its function: this is what I refer to as “physiosemiotics”. Regarding the first, we can generalize in treating the “channel” as a physical mechanism that causally connects states of an object also known as “sender” and states of an interpretant also known as “receiver”. Consider a snake (object) and a rabbit (receiver), which causally connect via audio-visual physical media, i.e., a “channel”. However, we cannot distinguish between channel and “message”, since the sign is part and parcel of the mechanism, such as a certain pattern of dimly discerned movement in the creaking grass. Hence, the sign is physical in turn, but only becomes causally operative via the interpretant. The interpretant responds to the sign in fulfilling a function, in case of the rabbit fleeing the snake to safeguard survival, which is a physical phenomenon in turn. In the physiosemiotic view, all aspects of information turn out to be physical. 

The concept of “function” dovetails with the pragmatist approach to meaning and knowledge that Peirce introduced in philosophy [24]: knowledge is what solves problems that an agent is facing. Whereas “meaning” is almost necessarily imbued with mentalist references, “function” does not inherently refer to epistemic subjects. I cannot delve into the philosophical intricacies here, beginning with properly interpreting Peirce’s original views (which tend towards a sort of panpsychism) and the contemporary discussion about function (Searle [25], for example, approaches function as being “observer dependent”). My view follows modern biosemiotics receptions of Peirce’s framework.

One important consequence of the physiosemiotics view is how we conceptualize the “noise” in the Shannon approach. Peirce starts out from the assumption of fundamental randomness of the world (his “tychism”) and asks how regularities arise (to which he often refers as “habits”). Considering the channel, we can conceptualize this as a correlation between states of the sender and receiver. Then, “noise” is an exogenous causal impact on the communication which is not correlated at all to the causal process connecting these two states and results in lowering their correlation. Once the meaning of the message becomes blurred, the resulting response of the receiver is becoming randomized (in Peircean terms, exogenous noise reduces the potential for formation of “habit” in sender-receiver interaction). 

Once noise becomes dominant, this means that both states become statistically independent. Then, referring to the receiver as system *X* and the sender as system *Y*, for the joint distribution *p* the condition $p_{\left(X,Y\right)}=p_{X}p_{Y}$ applies, such that the mutual information *I* between states of the two systems is zero:(1)$$I_{p}\left(X,Y\right)=\sum_{x,y}p(x,y)\log\frac{p(x,y)}{p(x)p(y)}=0$$

This can be expressed in Shannon entropies [26]:(2)$$I_{p}\left(X,Y\right)=H\left(X\right)+H\left(Y\right)-H\left(X,Y\right)$$

Or in terms of conditional entropies:(3)$$I_{p}(X,Y)=H\left(X\right)-H\left(X|Y\right)=H\left(Y\right)-H\left(Y|X\right)$$

That means that whereas the channel in the Shannon view distinguishes between message and channel, in the Peircean view the channel is the message (paraphrasing MacLuhan’s famous dictum, “the medium is the message”), and there are simply different degrees of correlation between the states of *X* and *Y*. In other words, there is no separation between “noise” and “message” on part of the receiver who does not know (different from the external observer) what “noise” is and what “message” is (to be able to do so he would already need to know the message). Further, (3) makes clear that degrees of correlation do not imply any causal direction, which is important when further exploring the physical aspects. These correlations constitute in precise terms what Peirce denotes as “habits”, and there would be no separation between “message” (somewhere existing in a Platonic domain as an undisturbed phenomenon) and “noise”. 

However, this raises an intricate issue: if we approach “habit” as mutual information, who is the interpreter of that information? Clearly, as stated, we implicitly posit a third party, that is, an observer who measures the statistical correlation between *X* and *Y*. Indeed, mutual information is a concept that is entirely independent from whether system *X* is capable of reflexively observing its correlation with *Y*. If we refer to physical entities in general, mutual information can indicate causal interdependence: Hence statistical inference, as in econometrics, is a way to identify causes. However, as long as we stay in the Shannon framework, we cannot subsume the causal force of the message under the causal mechanisms that work in the transmission process. If the observer infers a causal mechanism from mutual information, she presents a causal hypothesis that explains the correlation which is not part of the message. The question is whether there is a corresponding notion that does not imply reference to an external observer. This is the notion of function, if employed reflexively. That means, we do not refer to functions imposed exogenously on the correlated systems *X* and *Y*, but we assume that the mutual information feeds back on the underlying causal interdependencies in terms of sustaining or strengthening them, without intermediation by an observer.

Functional explanations state that a certain phenomenon exists because of is effects, such as in the standard example of the heart that has the function to pump blood through the organism [27,28]. Peirce already pointed out that this means to refer to a type of outcome as a causal force, and not just to a particular event as efficient cause [16]. This type of outcome could be achieved by various other means and relates to more general and more abstract frames of functions. The heart has the function to pump blood, the function of blood is to transport oxygen to all parts of the organism, this has the function to sustain metabolism, and so on. The debate about functionalism in the analytical philosophy of mind has shown that we can approach this in static terms but would need to take the entire reference frame of the specific function as a given, as in the example of the heart [29]. If we endogenize this reference frame, the only alternative seems to be evolutionary and selectionist explanations, commonly dubbed “etiological”. The paradigmatic approach is teleosemantics [30]. Teleosemantics comes close to the Peircean view [31] if we consider Peirce’s distinction between immediate, dynamic and final interpretants [16], with the final interpretant being the one which represents what we might refer to as fully revealed information, or “truth”, which in the context of functional explanations is tantamount to optimal or ideal functioning.

The evolutionary approach implies that information is a process concept, and not a “thing”: information grows, or, evolvability is what defines information [32]. The common critique directed at evolutionary explanations is that they would refer the value of information only to past selection events. However, this only applies in specific versions of evolutionary theory. If we include exaptation as a major evolutionary force, the notion of function includes the emergence of novelty independent from selection history [33]. That means that information is crucially dependent on context, such that any form of contextual change can change the function even radically. Most importantly, this also applies for a purely random variation of the original function, which in a different context may prove decisive for further change, even though in the moment at which it occurs it is without any function. Therefore, evolution is both past- and future-oriented: the future unfolding of the incipient function via ongoing actions of the interpretant creates a directedness of evolution, with the final interpretant as the vanishing point [34].

### 2.2. Autonomous Agents and the Physical Economy of Information

Now, if we approach semiosis in a dynamical form, as Peirce also did in distinguishing stages of evolving interpretants, we recognize that there is the possibility that functions may be fulfilled in diverse ways and with different degrees of proper performance: a rabbit may be too slow in processing the snake signs and running away. Signs may be erroneously interpreted (the snake is a mouse). Why does this matter? This is the point where economics comes into play. We only need to introduce the notion that information processing is costly. Indeed, this is the fundamental assumption in mainstream information economics which should be preserved in the otherwise dissenting view presented here. 

There are two aspects that loom large. The first is that transmitting, storing and managing information comes with a cost. The second is that there are costs of inadequate functioning. Both can be combined in the notion of “efficiency” since this relates inputs and outcomes. Efficiency refers to achieving the best possible functioning at the lowest costs, which is a core economic concept. Interestingly, this concept has been received in evolutionary and behavioural ecology, referring to measures that are not anthropomorphic, such as market prices as measure of cost, but which are physical in a most general sense, that is, referring to energetic costs of information processing, such as in the context of foraging, and the energetic gains for maintaining metabolism, survival and, ultimately, reproduction [35].

We can generalize these observations by introducing the concept of “autonomous agent” **AA** following Kauffman [19], which is more specific than the uses of the term in the AI literature in explicitly referring to metabolism. I define an autonomous agent by the following properties:An **AA** is an individual demarcated by physical boundaries between environment and internal systemic states. Environmental states are only accessible via “observations” (or: “measurements”).The **AA** actively reproduces these boundaries and pursues the goal to sustaining its existence through time, involving a distinction between “self” and “non-self” (e.g., the immune system).For achieving this goal, the **AA** operates a metabolism, i.e., manages flows of material and energetic throughputs to generate work. Observation is a kind of work.These operations require the physical processing of information generated by observations that is necessary for obtaining and processing the resources needed by the metabolism, and generally, proper functioning of the **AA**.

We can combine the notions of metabolism and information processing in the concept of “work”: we define **AA** as a physical entity that generates physical work that is directed towards the goal to maintain its existence, and that therefore is “autopoietic”, i.e., does not depend on goal assignments by an external agent. This distinguishes all kinds of devices from **AA**s. Yet, the general concept of work refers to both. This concept is complex and certainly crucial for economic information, since implicitly most economic uses of information relate to the capacity to work, such as in the production function. However, after the demise of the labour theory of value, economics has never produced a concise approach to “work”.

The relationship between work and information springs to the eye if we consider the distinction between the release of free energy via work and via dissipation as heat [36], pp. 31ff. Heat, by definition, means to “burn information”, in the sense of randomizing the movements of molecules in the environment. Work involves negentropic and hence informational states in two ways [34], p. 326ff, [37]: firstly, work is enabled via constraining the release of energy (think of positioning a wheel in a waterfall), and second, thereby work generates ordered changes of the environment (such as lifting a weight counteracting gravity). The capacity of an **AA** to generate work depends on its available free energy which is determined by the available part of its internal energy and external energy inflows. At the same time, the capacity to work is determined by the information stored in the **AA**, which is tantamount to the constraints on the release of free energy and can be measured in as Shannon information. This relationship has been already recognized by Robert Ayres [10], p. 37ff, in the simplest form of stating:$$H=\frac{A}{T_{0}}$$
with *H* the Shannon entropy stored in the **AA** and *A* its potential to generate work, i.e., its free energy. In recent research on the thermodynamics of information, this relationship has been explored much deeper and systematically (for the details on the following, see [26]). For non-equilibrium systems such as **AA**s, we can establish a direct connection to mutual information. However, we introduce an important change: the system *X_AA_* does not correlate directly with the environment *Y*, but with an observation *O*. Considering the fundamental equality between physical and informational entropy $S_{p}=kH\left(X\right)$, we can determine the change of physical entropy resulting from an observation as
(4)$$\Delta S_{O}=k\left(H\left(X|O\right)-H\left(X\right)\right)=-kI\left(X,O\right)$$
which allows one to determine the change of non-equilibrium free energy resulting from the observation:(5)$$\Delta A=-T\Delta S_{O}=kTI\left(X,O\right)$$

This establishes the basic relation between energy, work and information, in terms of the value of information as free energy gained from which work can be generated. 

This view defines the fundamental economics of **AA**s in purely physical terms, as an “autopoietic heat engine” (Figure 2). This diagram has been originally designed by Timothy Garrett [38] referring to economic systems in toto, but it directly applies to **AA**s as well.

The diagram describes what Kauffman refers to as a “thermodynamic work cycle”. The first important insight gained here is that we must neatly distinguish between two fundamentally different aspects of relations between the **AA** and the world: one is the world as “surroundings”, that is, a thermodynamic sink (*T* < *T_AA_*) without which the generation of work is physically impossible, following the laws of thermodynamics, and the other is that for sustaining its existence, the physical world is the “environment”, that is, the set of resources that are required by the **AA** metabolism. At a certain temperature *T_AA_* and pressure *p*, the boundary between **AA** and environment determines Δ*A*—the available energy in the environment that can be harnessed via the transfer of material and energy (apart from photosynthesis, mostly material) necessary to maintain the metabolism. This is exclusively determined by the internal **AA** operations, and hence depends on its information processing or “observation” in the previously established sense: the environment is physical, but at the same time semiotically determined, or, it is that part of the world that is functional in relation to the **AA**. In biological ecology, this is the “niche” [39]. The same physical world allows for a potentially unlimited number of “environments” also known as niches relating to various types and even individuals of **AA**. The only binding constraint is the general maximum availability of free energy flow per unit of time and space which could be harnessed for physical work, including the autopoietic kind and the activity of observation.

The next important insight is that autopoietic heat engines manifest a physical tendency to grow, depending on the efficiency of energetic transformations and of the work in relation to goal fulfilment. In detail, the inflow with rate *a* makes available energy accessible depending on internal parameters of transformational capacity *α* (which, most generally, can be conceived as embodied states of information). In addition, the physical transformations operate with a thermodynamic efficiency *ε* to produce work. We conceptualize work as being exclusively devoted to maintaining the **AA** (meaning, the **AA** does not fulfil any exogenously imposed function). Work changes the availability of energy in the environment. Hence, I assume, following Garrett, that this changes with the rate of work. This closes the autopoietic causal circuit. We conclude that the system will grow in the sense that the inflow with rate *a* will grow with a rate that is determined by the parameters*α* and *ε*:(6)$$\frac{da}{dt}=\alpha \frac{d\left(\Delta A\right)}{dt}=\alpha \varepsilon a$$
where we can define the product *α**ε* as the rate of return to autopoietic work.

Growth implies that more resources are consumed, such that in case of limited availability there are also limits to growth: a typical pattern emerging from such constraints is the logistic growth curve which is as widely applicably general characterization of growth processes in nature and the economy [40].

Growth is also essential when considering a population of **AA**s, since it drives competition over resources and hence creates the conditions for natural selection of **AA**s: if there is variation across **AA**s in exploiting resources, there is also variation in efficiency and hence growth. That means, there will be a change in the composition of the population, with a higher share of most efficient **AA**s at the carrying capacity. If we add the possibility of reproduction of **AA**s, we arrive at the simplest conceptual model of biological evolution. In this model, information as accumulating via natural selection would be directly referred to the capacity to harness energetic resources [41]. The question is how we can refer this to the general relationship between mutual information, entropy and energy established in (4) and (5).

## 3. The Physiosemiotic Approach to Semantic Information

In this section, I will present a more formal approach to physiosemiosis that builds on the important contribution of Kolchinsky and Wolpert [21] (KW). The KW approach directly ties up with the analysis of autonomous agents. However, KW explicitly exclude evolutionary considerations because they aim at constructing a general physical concept of semantic information and because they adopt the common critique of etiological explanations. In the second subsection, I will show how the KW approach can be plugged into the semiotic triad and thereby can be interpreted in the Peircean evolutionary framework. This results in a new physiosemiotic concept of semantic information.

### 3.1. Semantic Information and Thermodynamics

In the following, I present only the bare bones of the KW approach, referring the reader to the original contribution for more detail. I focus only on what KW call “stored information” and leave out the more complex discussion of “observed information”. This suffices to present the principles and fits to the previous discussion of **AA**s as embodying stored information that results from selection and determines the capacity to harness energy. 

The central concept is that of a viability function, which is measured in terms of negentropy at a fixed state space:(7)$$V\left(p_{X_{t}}\right):=-S\left(p_{X_{t}}\right)=\sum_{x_{t}}\left(p_{x_{t}}\right)\log\left(p_{x_{t}}\right)$$

In principle, this is a simple complexity measure: The reference point for assessing the complexity of an **AA** would be the purely random distribution of its constituent molecules in a constrained space. Clearly, that would also imply that there is no distinction between those possible distributions with high negentropy which have no autopoietic functional capacity and those which have. KW present a solution to that quandary. They distinguish between the system *X* and its environment Y, with the probabilistic distributions of respective random variables X and *Y*, *p_x_* and *p_y_*. They define “syntactic information” as the mere correlation of states between *Y* and *X* with the joint distribution *p_X,Y_* and the mutual information at time 0:(8)$$I_{p}\left(X_{0},Y_{0}\right)=\sum_{x_{0},y_{0}}p(x_{0},y_{0})\log\frac{p(x_{0},y_{0})}{p(x_{0})p(y_{0})}$$

I notice already here that in this formulation, there is not yet a distinction between the surroundings and the environment, i.e., the world as the set of hidden causes and the world as being informationally accessible for the **AA**, i.e. the world as being observed. This is crucial for my later reasoning, as we refer the viability function to either of the two when analysing **AA**-world relations [42]. 

Stored information can be measured as mutual information. However, for assigning informational value, this information must relate to the viability function. KW introduce the notion of “intervention”, which simply means scrambling the mutual information with the extreme of pure randomization, i.e., entire loss of any information. As we discussed previously, and as must be emphasized again, this suggests an external observer who defines and implements that intervention (by which the distinction between surrounding and environment is neutralized). 

KW arrive at a measure of the total value of stored information:(9)$$\Delta V_{tot}^{stored}:=V(p_{X_{t}})-V({\hat{p}}_{X_{t}}^{full})=S({\hat{p}}_{X_{t}}^{full})-S(p_{X_{t}})$$
where the roof indicates the scrambling, which is “full” when evaluating total value. However, what is most interesting is partial scrambling, which means to delete only a part of stored information. Then we can compare the different results for various scrambles and thereby assess the value of the scrambled part in terms of the different values of the viability function.

KW specify this intervention in terms of a coarse graining function ϕ(y). This function corresponds to the notion of “observation” introduced in the previous section. The interaction between environment and system is conceived as a communication channel in the sense that certain states of the environment induce certain states of the system, which is measured in terms of the conditional probability p(X|Y) such that the intervened channel would be written ${\hat{p}}_{X_{0}|Y_{0}}^{\phi }$.

We can then conceive the coarse graining function as defining which distinctions the system can make about the environment, resulting in a specific distribution of conditional probability:(10)$${\hat{p}}^{\phi }(x_{0}|y_{0})p\left(x_{0}|\phi (y_{0})\right):=\frac{\sum_{{y^{\prime }}_{0}:\phi \left({y^{\prime }}_{0}\right)=\phi (y_{0})}p(x_{0},{y^{\prime }}_{0})}{\sum_{{y^{\prime }}_{0}:\phi \left({y^{\prime }}_{0}\right)=\phi (y_{0})}p({y^{\prime }}_{0})}$$

KW use this formula to assess the effects of different coarse graining functions, with the extreme points of full scrambling where mutual information is zero and identity mapping. 

KW now define an optimum criterion. The original viability function is indeterminate regarding the value of the various distributions with same negentropy. However, if we run a series of interventions with varying coarse graining functions, we can separate those functions which affect the viability of the system and which do not. This allows one to define an optimum intervention as that coarse graining function which minimizes the amount of syntactic information while maintaining the same value of the viability function:(11)$${\hat{p}}_{X_{0},Y_{0}}^{opt}\in \underset{p^{\phi }:\phi \in \Phi }{\arg\min}I_{{\hat{p}}^{\phi }}(X_{0},Y_{0})\;s.t.\;V\left({\hat{p}}^{\phi }{}_{X_{t}}\right)=V\left(p_{X_{t}}\right)$$

KW can now distinguish syntactic information and semantic information in terms of the mutual information of the optimal intervention:(12)$$S_{stored}:=I_{{\hat{p}}^{opt}}(X_{0},Y_{0})$$

Further, KW define the efficiency of the system as
(13)$$\eta_{stored}:=\frac{S_{stored}}{I_{p}\left(X_{0},Y_{0}\right)}$$

I add for later discussion that this implies that highest efficiency indicates no slack in the information stored in the **AA**, and that this information is only about causally relevant mutual information. At the same time, however, KW point to the possibility of redundancy if there is more than one optimal intervention: that means that the **AA** would have alternatives with same viability value to choose from. 

Based on this, KW add the thermodynamic dimension. This is grounded in the Generalized Landauer Principle which states that any process that reduces the entropy of a system by *n* bits must at least export or can absorb at most $n\cdot k_{B}T\ln2$ of energy, coupled to an environment (heat bath) of temperature *T*. We can refer this to the notion of work as deployed by an **AA**: the **AA** expends work to gain information about the environment which costs at least $k_{B}T\ln2$ energy. If, as assumed previously, this work serves to harness energy, we end up with a simple measure of thermodynamic efficiency. Information that does not result in increasing energy inflows would be wasteful, and only information with a net gain would be functional. KW define the costs of acquiring new mutual information as:(14)$$W_{\min}=k_{B}T\ln2\cdot I_{p}(X_{0};Y_{0})$$

If we combine this perspective with the concept of viability value as defined by Equation 8, KW arrive at the cost/benefit ratio:(15)$$\kappa_{stored}=\frac{\Delta V_{tot}^{stored}}{I_{p}\left(X_{0};Y_{0}\right)}=\frac{S({\hat{p}}_{X_{t}}^{full})-S\left(p_{X_{t}}\right)}{I_{p}\left(X_{0};Y_{0}\right)}$$
called the “thermodynamic multiplier”. If this is larger than one, acquiring the information creates more energetic benefits than costs. Finally, KW relate the cost/benefit ratio to semantic efficiency:(16)$$\kappa_{stored}=\eta_{stored}\frac{\Delta V_{tot}^{stored}}{S_{stored}}$$
such that a lower semantic efficiency reduces the thermodynamic multiplier.

Now, consider the following problem. KW assume that the coarse graining function is modified exogenously by the observer. How can we envisage that this is generated endogenously by the **AA**? The only way is to assume an evolutionary process by which alternative ϕ(y) are generated as “mutations” which are the tested against the environment. There are two ways we can envisage that. The first is that the **AA** continuously tries out alternative ϕ(y), which is thermodynamically costly and involves risk of failure. Therefore, another option is “vicarious selection” of ϕ(y) [43], which would mean to run alternative versions internally against a model of the world and apply an internally represented value function that relates to the viability function. I will discuss this in the next section.

The other option is to analyse evolutionary processes on the population level: so far, we have only considered single **AA**s. Here, it is straightforward to relate the various measures proposed by KW to the general model of the autonomous agent. This model did not specify how the parameter *α* is determined, which reflects the capacity of the **AA** to exploit the available energy in the environment. This capacity corresponds to stored information in the KW approach. This implies that on the population level we consider variants of **AA**s with diverse capacities to grow, which implies that, over time, there will be a change in the composition of the **AA** population, such that the most efficient **AA**s will tend to dominate. 

That means, however, that we must refer the concept of information to the population level: even though stored information is a physical property of the individual **AA**, the information that is embodied in the **AA** is a population level phenomenon, causally rooted in the unfolding sequences of variation, selection and retention of variants. This leads us to reassess the notion of efficiency. As said, the maximum efficiency would imply that there is no more functionally neutral mutual information embodied: on the population level, that would correspond to the convergence of **AA** variants to one optimal type (or functionally equivalent forms). That would correspond to Fisher’s Fundamental Theorem on the population level: as a result of evolution, all variants would share the same information. However, this would imply that there is no more any variation available that could possibly generate a further growth of information.

This observation points to the general argument that maximum efficiency would minimize the capacity to evolve, which requires a potential of random variation [44]. In principle, this can be realized by preserving a gap between $S_{stored}$ and mutual information which can be activated in new coarse graining functions that would further increase semantic information. This would translate to retaining variance on the population level, thus ensuring evolvability. This important observation can be best elaborated in establishing the physiosemiotic framework.

### 3.2. The Physiosemiotic Triad

I will now plug the KW concepts into the physiosemiotic framework and build bridges to the general model of **AA**. I claim that the semiotic triad can be interpreted as a “unit” of semantic information, akin to the construct of Shannon information, and therefore substantially complementing and enhancing Shannon information. More specifically, the triad shows the fundamental structures of information embodied in **AA**s. Figure 3 shows the basic correspondences.

The semiotic object is the environment *Y*. This environment is semiotically constituted in the biological sense of a niche. That means, we do not directly refer to the “world” here, but the “surroundings” of the **AA**. The surroundings matter in two respects. First, as in Figure 1, they enable the physical generation of work in absorbing entropy export. Second, they represent the “unknown unknowns”, that is, in terms of statistical inference, hidden causes of any kind of correlations between *X* and *Y*. *X* is the interpretant. In combining the general model with KW, I posit that the interpretant realizes the function to maintain its existence, which, in KW, is captured by the viability function. 

Now, the crucial difference between the physiosemiotic approach and KW is that KW approach interventions as exogenous to arrive at precise definitions of their various concepts and measures. Specifically, the coarse graining function is varied by the external observer to measure that share of syntactic information, which is semantic information, i.e., distinguishing non-functional mutual information from functional. In the physiosemiotic framework, the coarse graining function is the sign or representamen. Signs emerge endogenously in the entire causal structure of semiosis, and their value is determined by the interpretant.

To catch this with the KW framework, we formulate the principle of bimodality: the actions by the **AA** in fulfilling its functions are determined in two modes. One is the channel in which causal impacts of the environment result in a specific correlation between states of environment and states of **AA**, which, however, is not yet reflected in causal assumptions, that is, informationally compressed, thus still retaining much non-functional correlations. Think of the rabbit and the snake again, in the moment when the rabbit perceives a movement and a sound, there are many other motions and sounds in the environment simultaneously impinging on its state (as formulated earlier, “noise” and “message” in the Shannon sense are merged). The coarse graining function would select certain segments. If, according to KW, the function minimizes costs, it would identify exactly those segments which identify a snake and thus trigger functional action, i.e., fleeing the scene. This is a sign of the presence of the snake.

As a result, and already analysed in von Uexkuell’s work of the “Umwelt” of animals [45], the environment is ultimately constituted by semiosis: snakes become objects (and not parts of snakes merged with a stone). The snake is real, but its ontological recognition is driven by semiosis. However, this conclusion only holds if we consider stored information S. This is the information resulting from a series of interventions that have identified the optimum, i.e., the least costly state of information. In Peirce’s framework, this is the final interpretant. That means, as stated previously, we must understand the semiotic triad as manifesting a process, not a state. How can we achieve that without exploding the frame? 

My suggestion is that we must change our fundamental assumptions about the ontology of causation, following and extending on Peirce’s ideas about the interplay of efficient and final causes in constituting the world as an ordered flow of phenomena [46]. This has been extensively elaborated by Terrence Deacon [34], building on earlier contributions in the biosemiotics literature, such as Stan Salthe’s work [47]. That means that we go back to the original Aristotelian distinctions [48]. This step is surprisingly simple if we ask the same question as originally asked by Aristotle: what does it mean to search for causes of observed phenomena, and in our case, more specifically, if we ask for causes of phenomena displayed by **AA**s? Clearly, considering the rabbit again, it does not suffice to refer to the efficient causal impact of physical media on its behaviour. The first step is that we must refer to the sign that triggers the fleeing response: coarse graining corresponds to the Aristotelian notion of a formal cause, as the sign categorizes the various efficient-causal impacts, and accordingly a type of efficient causes (and not their tokens) causes a type of action, i.e., fleeing. Yet, we must still ask why this specific relationship between sign and response realizes, and recurrently so (a “habit” in Peircean parlance). Here we must refer to function, and hence an Aristotelian final cause.

As in the previous discussion of function, ultimately, we must explain the function, which requires reference to a selection context. However, considering the case of exaptation, the final cause would not necessarily relate to a past selection history, but to the emergence of a new function “in being”, i.e., in the sense of the future-directedness [49]. This is much emphasized by Deacon. Accordingly, we can directly apply the concept of final cause on physiosemiosis. This cannot be grasped in the KW formalism but is implicit to the notion of optimal intervention as the final state of a series, i.e., we can think in terms of approximating interventions. In fact, this is implicit to the KW measure of efficiency, since this refers to that part of mutual information at *t_0_* which is sufficient for viability at time *τ.*

In introducing the physiosemiotics approach I emphasized the pivotal role of the interpretant. However, we now realize that final causes and efficient causes relate to each other in the evolutionary emergence of information: the world is not just a “subjective” construct by the interpretant, but what is constructed physiosemiotically as the environment relates to hidden causes in the world. As argued by KW, the stored information eventually is information about causes, and more specifically, efficient causes. That means that the series of interventions ultimately results in a picture of the world that transcends mere mutual information but embodies causal forces. This is exactly Peirce’s view, moving from a mere stochastic world to a world of “habits”, i.e., causal regularities. This matches with classical positions in evolutionary epistemology, which claim that only evolutionary processes can generate true knowledge.

Finally, we must pick up the thread left loose in the previous section, where I argued that the coarse graining functions evolve on the population level (Figure 4). This idea has been adumbrated by Peirce who refers the notion of habit to “communities” [50]. In the biosemiotic context, this means that signs acquire meaning on the species level, whereas the individual level is that of “meaningless” variation. If we adopt this interpretation, we can conclude that signs and function are determined by functional causes on the population level, whereas efficient causes operate on the individual level. By implication, signs and the related species term assume existence in the ontological sense: this reflects Peirce’s vigorous rejection of metaphysical nominalism (the complex discussion of the species concept in biology is highly relevant here [51]).

## 4. Consequences for Theorizing the Economic Agent: The Case of Neuroeconomics

In conclusion, I want to explore some consequences for economics proper. I concentrate on the question: what does it mean to treat economic agents as autonomous agents? Economics approaches agency as choice based on the processing of information. Is the approach suggested here compatible with the conventional economic model, given the centrality of concepts such as efficiency, or is it an alternative view?

Information plays a crucial role in the standard conception of economic agent, which vacillates between various versions with one extreme pole of assuming a perfectly and completely informed rational agent in classical versions of general equilibrium theory and the other extreme of “zero intelligence agents” in some models of artificial markets [7]. The shades in between are mainly defined via two basic versions: one is assuming simply quantitatively constrained information, such as its unequal distribution across agents, and the other is adding cognitive limitations, as in behavioural economics, which jeopardize processing even of available information.

If we approach economic agents as **AA**, we arrive at a different picture. I cannot explore the details here due to limited space, but the basics [52]. The starting point is to approach the economic agent as a physical entity; for simplicity, I reduce this to the human brain in the first place, although in the full analysis this must be widened to the body and the conjugated entities, such as cultural artefacts. The focus on the brain allows me to connect directly with the recently emerging field of neuroeconomics. Which insights can be gained for neuroeconomics once we approach the brain as the information processor of human **AA**s? The main insight is that the brain is an evolutionary physiosemiotic system. 

In my following discussion, I concentrate on the narrow field of neuroeconomics, and not economics in general, as this would open too many topics at the same time. I look at one of the dominant paradigms of neuroeconomics, the so called “good-based model” [53]. This is the clearest version of a general approach in neuroeconomics which aims at integrating neurosciences and economics on the basis of the established neoclassical model of choice, i.e., employing concepts such as utility function and optimization in building an embodied theory of subjective value. This model assumes that choice is based on a specific status of information, which refers to states of the world and assigned probabilities, but otherwise strictly follows the formal structure of the economic model. One important consequence is the systematic distinction between informational states, choice and action in a sequence that is linear and does not allow for any feedbacks from current action to previous stages of the decision process.

Recently, an alternative view has been formulated which matches with some other versions of the basic neuroeconomic model [54], such as the drift-diffusion model [55]. This view approaches the brain as an evolutionary system—a perspective that we already introduced in the previous section. This view is anticipated in the long-time neglected work by Friedrich Hayek, “The Sensory Order” [56], where Hayek approaches the brain as a homeostatic system, directly referring to the emerging general systems theory of his time. In the new approaches, the basic idea is that the brain constantly generates contextualized and competing action plans which are directly revised and evaluated during the flow of action [57,58,59]. This operates according to a selectionist logic, such that eventually one action plan will end up as determining the next action. 

The important point is that this alternative model is explicitly grounded in phylogenetic considerations, referring to the general model of foraging in an uncertain environment, and thus directly tying up with the general model of **AA**. That means that there is a necessary connection between the requirements of harnessing resources and the way the physical structure of information processing in the brain evolved phylogenetically. This view has been most systematically elaborated in Cisek’s “affordance competition hypothesis” [60,61]. The adaptationist argument refers to the general scenario of foraging in which the human scavenger is constantly roaming an environment, facing two challenges. The first is that she never knows which directions end up with better action outcomes, which is only ascertained once action is taken, and hence other options are given up. It is crucial to recognize that this relates to a fundamental problem in the theory of information: if an agent must choose between different options that are uncertain, it does not know how to value the lost alternatives. Even assigning probabilities does not help, because the state space as such is partially unknown. This problem is leveraged when considering the multi-dimensionality of uncertainty, most basically, as the second challenge, the uncertain presence of predators and other threats to survival. Therefore, the human **AA** operates on the basis of “vicarious selection”, i.e., the internal evolutionary selection of action plans while acting. A core determinant is the capacity to interpret environmental cues, i.e., signs. This results in representing the environment as a set of affordances for action, mediated by the signs.

Accordingly, we can employ the physiosemiotic framework of analysing the **AA** to describe the economic agent. The economic aspect is defined by the energetics of information processing, which in turn determine the universal selective forces that impact on the **AA**. This can be regarded as a radically materialist or physicalist view of the economic agent, which will immediately invite the straightforward critique that many actions, even in the phylogenetically older states, do not involve energetics as in the foraging scenario, and that these become irrelevant once energetic constraints are no more binding in most types of actions, as in affluent modern societies. However, this critique suffers from a misunderstanding. The phylogenetic argument shows why fundamental structures of economic agency have evolved as a result of the energetics and economics of information processing [62]. These structures apply for all kinds of behaviour, whether explicitly directed at harnessing energy or not. This has immediate consequences for choosing among alternative models of economic agency: we conclude that the goods-based model, vindicating the standard neoclassical model, should be rejected in favour of models that integrate information processing, choice and action.

Recently, Karl Friston and collaborators [63,64] developed a “free energy theory of the brain” which is strongly compatible with the view developed here and offers a formal venue to combine the evolutionary approach with the thermodynamics of information as sketched in the previous section. However, they use statistical thermodynamics only in the formal sense, i.e., the eschew direct references to energetic considerations. Yet, their approach is relevant here because a core hypothesis is that the brain minimizes variational free energy in the sense of minimizing “surprise”, thus extending and improving on the rich literature on the “Bayesian brain” [65]. We can refer this to the KW notion of efficiency. The theoretical boundary of the most efficient state of embodied information would be one in which the “surplus” of mutual information about stored information is minimal. However, the free energy framework differs from the KW framework in regard to one crucial aspect that I already highlighted previously: the “observer” is endogenous, i.e., the **AA** itself, and there is no direct relationship between the external world and inner states of the **AA**. 

The **AA** connects via the world via sensory inputs and via its actions which relate to the former in two ways: first, actions respond to sensory inputs and generate sensory inputs, and second, actions select the conditions under which sensory inputs are produced. This is called “active inference”—a view long presaged by the developmental psychologist Jean Piaget, a contemporary of Hayek who also received General Systems theory [66]. As Friston and collaborators have recently elucidated [42], this view is “pragmatist” in a principled and radical way: models of the world are embodied in the brain, and they are enacted via action. Now, if one approaches the information generating process in statistical terms, this means to implement a form of Bayesian learning, which, however, is different from most “Bayesian brain” approaches because it is non-representational: distinguishing between two types of statistical models, the recognition density and the generative models, the brain does not aim at producing an accurate representation of the world, but at reproducing an action pattern that strives to minimize surprise. The connection to the KW approach is straightforward here, because minimizing surprise means to minimize the entropy of the state of the organism, which, however, is now seen as embodying the observations that are parametrized via the **AA**’s own actions. This can be formally treated via the free energy principle.

We can relate the enacted generative models of the world to the signs in the physiosemiotics approach. Signs are actively inferred from a “noisy” world, and they motivate actions which result in the structuration of the world: the environment or the niche is that generative model. The “world” is semiotically constructed in which humans strive to minimize “surprise”.

The free energy approach implies a radical reformulation of economic agency since the concept of “utility” would be discarded as a general medium of optimization and substituted by free energy minimization, or, even more generally, safeguarding the low entropic states of the human **AA**, in the KW sense [67]. This is close to the view of the so-called “old institutionalists”, such as Thorstein Veblen, that human behaviour must be analysed in terms of “habits”, and not in terms of rational choice [68]. This seems to suggest, however, that habit formation and optimization are opposing principles. The free energy approach shows that this is not a contradiction, as the habits emerge from a continuous effort to impose order on the world, hence aiming at minimizing surprise in Friston’s sense. The institutionalists were informed by Peirce: Peirce’s notion of finality is much closer to the notion of optimization than it seems. This opens the vista to a reconciliation of hitherto radically opposing positions in economics.

Further, if we look at the dynamic aspects of behaviour, the free energy approach has the important implication that choice is strongly contextualized. This is also emphasized in the evolutionary approaches to action generation: if the flow of actions aims at minimizing surprise, there is a continuous evaluation of contexts and implied goals, thus allowing for instantaneous revisions of preference orderings [69]. What counts is whether the flow of actions results in a stable and sustainable pattern, especially when it comes to social interaction, and not to maximize a de-contextualized, fixed and stable utility function over the space of goods. This pattern is embodied and enacted semiotically, such that we cannot reduce it to internal states of the brain [70].

Hence, there is no neat analytical boundary between the agent and the external world, since both are deeply semiotically integrated. Accordingly, we must radically rethink the concept of information: there is no flow of information from world to agent which is then processed and results in choices, but information is created by the actions of the agent. This actor-centredness of the concept of information transpires from the foundational work on thermodynamic information that I presented in this paper.

## Figures and Tables

**Figure 1 entropy-23-00277-f001:**
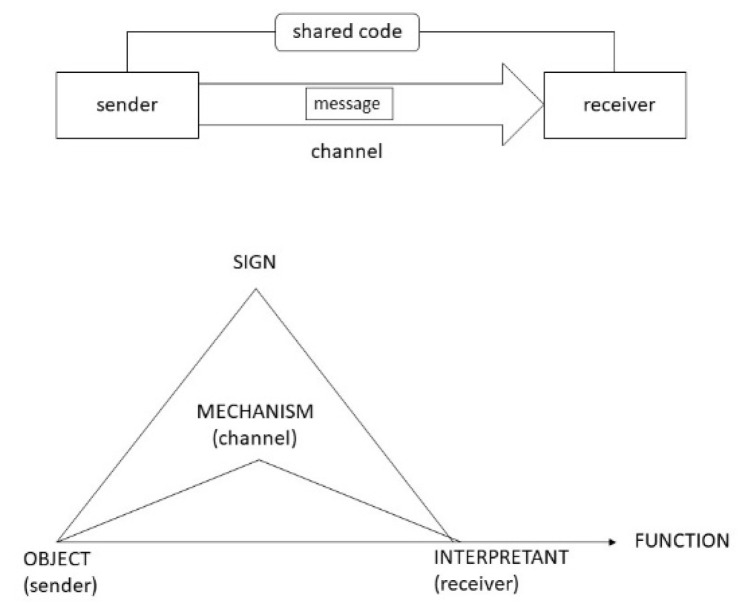
Comparison between the Shannon information theory and semiotics.

**Figure 2 entropy-23-00277-f002:**
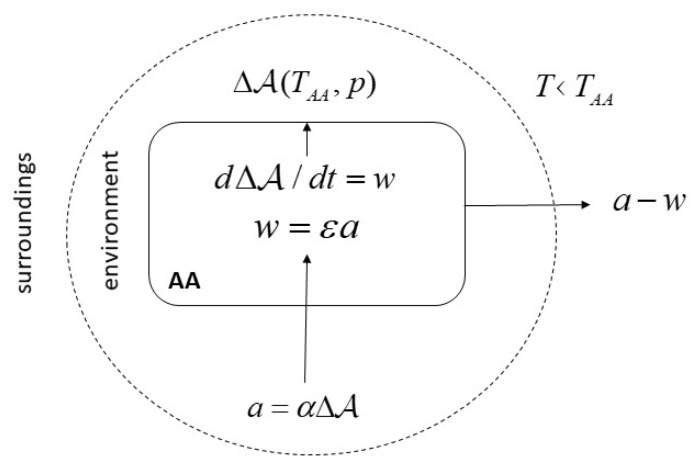
Autonomous agents as autopoietic heat engines (modified after Garrett 2011)

**Figure 3 entropy-23-00277-f003:**
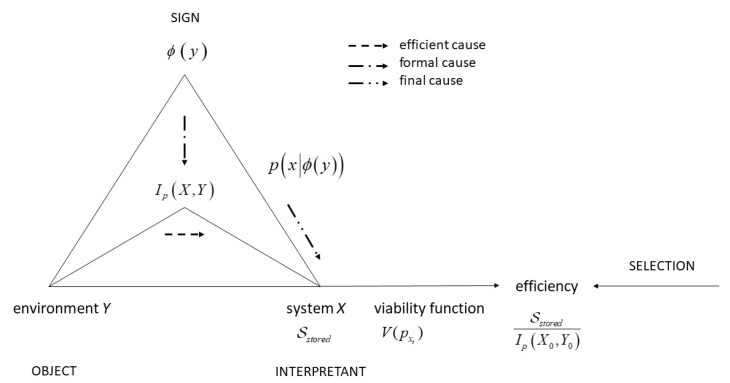
The Kolchinsky and Wolpert (KW) approach plugged into the physiosemiotics triad.

**Figure 4 entropy-23-00277-f004:**
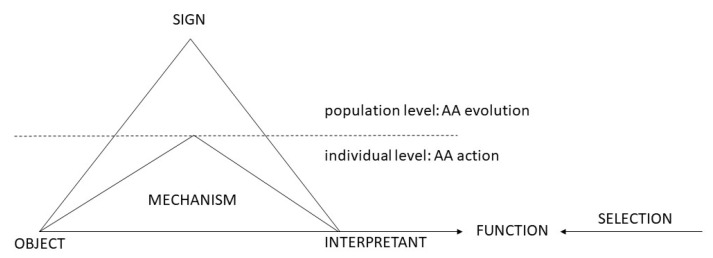
Physiosemiosis and evolution.

## Data Availability

No new data were created or analyzed in this study. Data sharing is not applicable to this article.

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
