# Peer review of "The Natural Philosophy of Economic Information: Autonomous Agents and Physiosemiosis"

_entropy, 2021, doi:10.3390/e23030277_

Round 1

Reviewer 1 Report

Specific questions to the author:

  1. “…approach economic agents as AA - for simplicity, I reduce this on the human brain in the first place”

This is unnecessary reductionism – it completely takes out the social environment from the picture as if a brain arises in vacuum. We had enough of absurd reductionism in the mainstream economic theorizing and such a narrow way of defining economic agents is not helpful. You already emphasized “the role of culture in framing, motivating and driving  choice.”

  1. “The free energy approach implies a radical reformulation of economic agency since the concept of ‘utility’ would be discarded as a general medium of optimization and substituted by free energy minimization, or, even more generally, safeguarding the low entropic states of the human AA”.

Again, this is replacing one abstract reductionism – utility – with another - free energy minimization. This may be a “radical reformulation of economic agency” in the context of mainstream economics, but from the point of view of heterodox theorizing, it is yet another unproductive description of complex human agency.

  1. “that human behavior must be analyzed in terms of ‘habits’ and not in terms of rational choice. The free energy approach shows that this is not a contradiction, as the habits emerge from a process of optimization. If we put this in the semiotic framework, this can be empirically specified in the role of culture in framing, motivating and driving ”

I do not see how the institutionalist concept of “habits” could correspond to “free energy minimization”. This should be explained in detail. Besides, habits are nowhere taken to be optimal – whatever process that humans agents use in problem-solving or in responding to external challenges is at best satisficing and never optimizing, as Herbert Simon pointed in his 1966 book.

  1. “…culture is a universal mode of existence, as a deeply semiotically constructed ‘world’ in which humans minimize ‘surprise’. ….

“If the flow of actions aims at minimizing surprise, “ ..   “minimizing surprise means to minimize the entropy of the state of the organism”

This needs explanation and clarification. Institutions (as an element of culture) do reduce the uncertianty of expectations in the context of human social interactions, but this is not equal to “minimize ‘surprise’”. Much less is clear, in the economic agency context, what does mean that “the flow of actions aims at minimizing surprise”  and how this equates with the entropy of the state of the organism? 

Author Response

Reviewer 1 concentrates on section 5.  My response is as follows.

  1. Now I write: "The starting point is to approach the economic agent as a physical entity; for simplicity, I reduce this on the human brain in the first place, although this must be widened in the full analysis to the body and the conjugated entities, such as cultural artefacts." Indeed, I agree with the reviewer, but my position, developed extensively elsewhere, is externalism and hence a form of 'materialism'. This is not reductionism, but emergentism. There is no space to discuss this here, why I also say "for simplicity". Hope the rephrasing helps.
  2. This is an attack in which the reviewer seems to enforce her/his view of 'heterodoxy' on my own position. I claim the freedom to express my own view. Further, it must be emphasized that I explicitly refer to neuroeconomics, not 'economics' in general. That is, I must take the state of the art in neuroeconomics as a reference, and not any other kind of heterodox thinking in economics. This is what I clearly say. In that specific context, it makes a huge difference to move away from 'utility'. However, I understand that I do not communicate this clearly, thus inviting criticism. I changed the title of the subsection to avoid this misunderstanding, and I added one sentence (fourth paragraph): "In my following discussion, I concentrate on the narrow field of neuroeconomics, and not economics in general, as this would open too many topics at the same time."
  3. The reviewer is right on this point, as far as the common use of the term 'habit' is concerned. I changed the paragraph. However, I make clear that if one goes back to Peirce, this is not exactly true: Peirce's notion of finality of course relates to a trend of 'optimizing' (such as approximating truth). To indicate the trend, I now use expressions such as 'striving for minimizing surprise'. Well, that's what the entire section 3 shows. Further, as Jim Wible's extensive work on Peirce and economics has shown, Peirce himself appreciated the emerging neoclassical thinking of his times. But these issues of history of ideas should not mess up these final paragraphs.
  4. To avoid these debates, I deleted the sentence on culture and hence any implicit reference to 'institutions'.

Reviewer 2 Report

The paper is written clearly demonstrating an very good experience of the author in this research domain. However, the author focuses on "providing the basis for a physical view of information in economics" introducing the idea of "physiosemiotics". That rises the question of how to understand information in economics when today we face the exponential development of the knowledge economy, behavioral economics, and the theory of knowledge-based firm? The author should expend his view from information toward knowledge and from economics based on tangible resources (i.e. physical objects) toward economics based on intangible resources (i.e. non-physical resources).

Author Response

I think the reviewer's point is perfectly legitimate, but would explode the length of the paper. I added a sentence right in the first paragraph to clarify the limitations of the paper, distinguishing between 'microfoundations' and 'macro-phenomena' such as the knowledge economy. My only concern is the former. I hope that this is an acceptable compromise.